# Influences of CNT Dispersion Methods, W/C Ratios, and Concrete Constituents on Piezoelectric Properties of CNT-Modified Smart Cementitious Materials

**DOI:** 10.3390/s23052602

**Published:** 2023-02-27

**Authors:** Tofatun Jannat, Ying Huang, Zhi Zhou, Dawei Zhang

**Affiliations:** 1Department of Civil, Construction and Environmental Engineering, North Dakota State University, Fargo, ND 58105, USA; 2School of Civil Engineering and Architecture, Hainan University, Haikou 570228, China

**Keywords:** smart concrete, piezoelectricity, carbon nanotubes, dispersion method, water-cement ratio, concrete constituent

## Abstract

In order to achieve effective monitoring of concrete structures for sound structural health, the addition of carbon nanotubes (CNTs) into cementitious materials offers a promising solution for fabricating CNT-modified smart concrete with self-sensing ability. This study investigated the influences of CNT dispersion method, water/cement (W/C) ratio, and concrete constituents on the piezoelectric properties of CNT-modified cementitious materials. Three CNT dispersion methods (direct mixing, sodium dodecyl benzenesulfonate (NaDDBS) and carboxymethyl cellulose (CMC) surface treatment), three W/C ratios (0.4, 0.5, and 0.6), and three concrete constituent compositions (pure cement, cement/sand, and cement/sand/coarse aggregate) were considered. The experimental results showed that CNT-modified cementitious materials with CMC surface treatment had valid and consistent piezoelectric responses to external loading. The piezoelectric sensitivity improved significantly with increased W/C ratio and reduced progressively with the addition of sand and coarse aggregates.

## 1. Introduction

As the most widely used construction material, concrete has taken our civilization forwards for centuries. However, concrete structures may be subjected to various environmental threats including erosion, impact forces, and harsh weather conditions. Excessive exposure to such adverse environmental conditions could induce cracking and delamination of concrete, leading to potential long-term safety concerns. Thus, close monitoring of concrete structures in these environments is needed to prevent catastrophic failure [1,2].

The most popular methods to inspect of concrete damage and deformations include visual inspection and the usage of attached or embedded sensors [3,4,5]. Although visual inspection is a cost-effective inspection method, it represents a tremendous workload due to the large size and complexity of most civil infrastructures. In addition, the results of visual inspection are sometimes unreliable given that most instances of concrete damage occur on the inaccessible parts of the structure [2,6,7]. For attached or embedded sensors, there are several types of sensing technologies available to monitor the conditions of concrete structures, such as resistance strain gauges, shape memory alloys, and fiber optical grating sensors. However, the attached or embedded sensors also have limitations such as challenges in compatibility with concrete and local measurements [2,6,8]. The needs for structural health assessment of civil infrastructures have necessitated research on the development of real-time and in situ monitoring techniques. Such techniques should allow the system to monitor its structural integrity while the infrastructure is in service, and the monitoring can be performed throughout the whole service life of the infrastructure.

Smart materials (e.g., smart cementitious materials) are intelligent systems with advanced properties such as shape memory or self-sensing [9,10]. Smart cementitious materials are usually fabricated by adding functional fillers (biological materials, crystals, ceramics, polymers, nanomaterials, etc.) into the cement matrix [11,12] to achieve the self-sensing purpose. The functional fillers usually take advantage of the piezoelectric effect. This is defined as a linear electromechanical interaction between the mechanical and electrical state in which electric charge is accumulated in response to applied mechanical stress in a crystalline material with no inversion symmetry [13,14]. With the addition of such functional fillers, the stress information of the cementitious material can be derived from the electrical signal measured in the matrix [15,16]. The piezoelectric effect is a dynamic process because the stress is proportional to the electrical signal and the signal conversion is a reversible process [13]. Smart cementitious materials retain superior precision and high sensitivity even when the material is under a high stress state [16]. Therefore, the development of smart cementitious materials with intrinsic piezoresistive or piezoelectric self-sensing capability is of great importance for monitoring concrete structures in real time [17].

Among the various different types of functional fillers, carbon nanotubes (CNT) are recognized as one of the promising functional fillers that enable cementitious materials to achieve self-sensing properties [16,18,19]. The surface piezoelectric effect under a non-uniform strain was found in multi-walled CNTs using atomic force microscopy [20]. In addition, as CNTs have remarkably high aspect ratios (hundreds to several thousand) and large surface area to volume ratios, exceptional high tensile strength (30–50 GPa) and elasticity (1.0–1.5 TPa), the addition of CNTs has been investigated as an approach to enhance the mechanical properties of cementitious materials [21,22,23,24].

Previous studies showed that achieving satisfactory sensing properties in CNT-modified smart concrete is highly dependent on the proper dispersion of CNTs in the cementitious matrix. In other words, CNTs need to be well-dispersed in a cement matrix to form an extensive conductive network inside the concrete to achieve consistent sensing properties [25]. However, CNTs tend to form into CNT clusters due to the considerable Van der Waals forces between them [26,27]. To reduce the size of the CNT clusters and improve their dispersion, the most prevailing CNT dispersion method is the mechanical stirring such as direct mixing and sonication [28]. The mechanical stirring approach is simple to apply, but it has been found insufficient to disperse CNTs uniformly in cement mortar. Therefore, CNT functionalization methods using various dispersing agents such as sodium dodecyl sulfate (SDS) and sodium dodecyl benzenesulfonate (NaDDBS) have been investigated to improve the CNT dispersion [25,29]. A suitable concentration of such dispersing agents may improve CNT dispersion [30,31] while at the same time, preserving or improving the physical and chemical properties of the cement mortar [32,33].However, the existing findings concerning surfactants on CNT dispersion modification vary significantly among different studies, showing the lack of consistency [34]. In addition, some dispersion agents may induce negative influences on cement hydration or void density [35,36]. Recently, a different CNT surface modification method using carboxymethyl cellulose (CMC) has been investigated and showed promising effectiveness in improving CNT dispersion consistently while increasing the piezoelectric sensitivity in cement mortar [37]. However, there is no investigation yet on the effectiveness of this new surface modification in concrete which has sand and aggregate in addition to cement mortar.

In addition to the effectiveness of dispersion, a good self-sensing property of smart concrete is also influenced by a variety of other factors such as the water/cement ratio and concrete constituents. The water/cement (W/C) ratio, a universally influential factor for most of the properties of cementitious materials, can change the deformation capacity of CNT-modified cementitious materials and CNT dispersion in the cement-based matrix [32,33]. Higher W/C ratios may result in higher electrical resistance [38] or better CNT dispersion for CNT-modified cement mortar, and thus, a potentially better piezoelectric sensitivity. Additionally, most of the existing studies of CNT-modified cementitious materials only investigated the influences of CNT modifications in cement paste without sand or aggregates [37], while in practice, concrete always has fine and coarse aggregates in addition to cement and water. The influences of concrete constituents such as sand and aggregates on CNT-modified smart concrete and the CNT dispersion effectiveness have not yet been investigated.

In this paper, the influences of the dispersion method, W/C ratio, and concrete constituents on the piezoelectric response of CNT-modified smart concrete are systematically investigated. Three different dispersion methods (direct mixing, NaDDBS, and CMC surface treatments), three W/C ratios (0.4, 0.5, and 0.6), and three concrete constituent compositions (pure cement, cement–sand, and cement–sand–coarse aggregate) were considered and experimentally tested. This paper will add to the fundamental understanding of CNT-modified smart cementitious materials with piezoelectric behavior for the purpose of self-sensing applications.

## 2. Materials and Methods

### 2.1. Materials

Since previous research showed that the multi-walled CNTs (MWCNTs) have superior electrical conductivity compared with the single-walled CNTs, the multi-walled CNTs (supplied by Skyspring Nanomaterial Inc., Houston, TX, USA) were adopted as the functional fillers in cement. The detailed specifications of the CNTs are listed in Table 1. The concentration of CNTs throughout this study was set at 0.1% by the weight of cement since it was reported as the optimal CNT concentration for cementitious materials to obtain a considerable electrical performance [37]. Regarding the dispersion agents, NaDDBS and CMC were provided by Sigma-Aldrich Co., St. Louis, MO, USA. According to the specifications from the supplier, the NaDDBS had an average molecular weight of 288.38 g/mol and a chemical formula of CH_3_(CH_2_)_11_OSO_3_Na. The molecular weight of the CMC used was around 90,000 g/mol and its chemical formula was CH_3_(CH_2_)_11_C_6_H_4_SO_3_Na. The matrix material and aggregates were purchased from The Quikrete Companies, Atlanta, GA, USA. The matrix material was Type 1 Portland cement which met ASTM C387 for compressive strength requirements. In accordance with the ASTM C33 standard, all-purpose sand with a diameter range from 0.3 mm to 2.36 mm was used as the fine aggregate, and gravels with an approximate diameter of 9 mm were applied as the coarse aggregate. Due to the fact that this study focuses on the investigation of piezoelectric properties of cementitious materials, no superplasticizer additives were applied in this study to avoid additional influencing factors and potential interaction among the plasticizers, surfactants and CNTs.

### 2.2. CNT Dispersion Methods

CNT-modified smart cementitious materials were prepared using three different dispersing methods, including mechanical stirring, and surface treatments using NaDDBS and CMC. Regarding the mechanical stirring, the direct mixing method was adopted in this study due to its low cost and ease of application [39,40]. The direct mixing did not involve any CNT treatment. 0.2 g CNT was added directly into 120 mL of water while the solution was mixed with a stirring bar on a magnetism stirrer. The stirring speed was 1600 rpm. The CNT–water suspension attained homogenization after 15 min of stirring. Then, 200 g cement was added into the CNT–water suspension, resulting in a 0.1% CNT concentration and a 0.6 W/C ratio. Thus, it is worth noting that the water used for CNT dispersion was included in the W/C ratio. To investigate the influences of the W/C ratio on the sensing properties of CNT-modified smart concrete, three W/C ratios (0.4, 0.5, and 0.6) were considered in this study as they are the typical W/C ratios used in the field. Similar procedures were followed for preparing the 0.5 and 0.4 W/C ratios with the water amounts being 100 mL and 80 mL, respectively.

For the NaDDBS surface treatment, a critical micelle concentration of NaDDBS in water of 1.4 × 10^−2^ mol/L (approximately equal to 0.4875% by weight) was set as the input surfactant concentration [35]. First, 1.17g of NaDDBS was mixed with 120 mL of water using a magnetism stirrer for 15 min. While stirring, 0.2 g of CNT was added to the aqueous solution followed by 2 h of sonication to ensure full interaction between the NaDDBS and CNTs. Then, the NaDDBS/CNT solution was mixed with 200 g of cement. A previous study found that air bubbles can appear in the cement paste [35] with NaDDBS addition. Therefore, 0.25% of defoamer (by volume) was utilized to decrease the air bubbles in the CNT-filled cement pastes. The defoamer was Tributyl phosphate supplied by Sigma-Aldrich Co., USA.

Regarding the CMC surface treatment method, a previous investigation by the authors determined that 0.5% CMC was the optimal percentage to ensure piezoelectric sensitivity of cement paste. Thus, 1.2 g of dry CMC was gently added into 120 mL of water, since CMC tends to clump in water owing to its high-water absorption and retention. The CMC solution was mixed on a magnetism stirrer for up to 30 min until the CMC was free of clumps and completely dissolved into the water. After mixing with 0.2 g of CNTs, the CMC/CNT solution was transferred to a 50 mL test tube and placed on a tube rotator for at least 72 h to ensure a proper coating of CMC on the CNTs. Then, 400 g of cement was added into the solution. The key dispersion procedures of NaDDBS and CMC surface treatments are shown in Figure 1.

### 2.3. Samples Preparation

In addition, to investigate the influence of concrete constituents, three cementitious composites were studied including pure cement paste, cement mortar, and concrete with cement, sand, and coarse aggregate. To prepare the cement mortar, 400 g of sand was added into 200 g of the cement pastes which were prepared in the last section, yielding a 1:2 design mix ratio of cement and sand. To prepare the concrete with cement, sand, and coarse aggregate, the well-prepared cement paste was mixed with 400 g of sand plus 600 g of coarse aggregate by hand using a trowel, resulting in a 1:2:3 design mix ratio of cement, sand, and coarse aggregates. After all the materials were thoroughly mixed, the pastes were placed into molds to make concrete cubes. The edge length of a concrete cube was 50.8 mm, as shown in Figure 2a, and the other sample configurations and manufacturing procedures are based on ASTM C109. The samples were cured in the molds for 24 h at room temperature (22 °C ± 2 °C). An exception was that all the cement pastes and concrete samples with 0.5% CMC needed 30 h of sitting time in the molds to stay intact. The electrical wires were placed 12.7 mm deep and 12.7 mm apart from each other in each sample prior to the solidification of the concrete mixture. The samples were demolded and placed in water for 7 days to cure followed by 10 days of air drying at room temperature. Figure 2b demonstrates a ready-mixed sample.

Table 2 presents the testing matrix. For each testing group, three identical samples were fabricated to be statistically valid. Group A was used as a control group to confirm that the cement mortar without any functional fillers could not exhibit any piezoelectrical sensitivity. As no piezoelectrical effect is expected from either cement mortar or concrete without functional fillers, only three samples with a W/C ratio of 0.6 were fabricated in the control group. As previous studies investigated how the W/C ratio influences the piezoelectrical effect on pure cement paste using a direct mixing method [32,41,42], as discussed in Section 1, Group B (0.6 W/C), C (0.5 W/C), and D (0.4 W/C) were designed to investigate the influences of the W/C ratio for smart cement mortar using the direct mixing method. If significant piezoelectrical effects were found in these samples during the dynamic loading tests, more samples for the three different W/C ratios for smart concrete with aggregates would be further fabricated. Since the CMC surface treatment method is a relatively new CNT dispersion method, there is a lack of related investigations regarding the piezoelectric effect of CNT-modified smart cementitious materials. Thus, in this study, samples with three W/C ratios (0.6, 0.5 and 0.4) and two concrete constituents (pure cement and cement mortars) were prepared with the CMC surface treatment method. Groups E to G and H to J were designed to investigate the influence of the W/C ratio on CMC surface-treated CNT-modified cement paste and cement mortar, respectively. The optimal W/C ratio from Group E to J was used to fabricate the sample Group K for CMC-treated CNT-modified concrete with cement, sand, and aggregate to investigate the influences of course aggregate on the piezoelectrical effect. To investigate the influence from different dispersing method on the piezoelectrical effect, Groups L to N were fabricated as NaDDBS-treated CNT-modified cement paste, cement mortar, and concrete with coarse aggregate.

### 2.4. Experimental Setup

The prepared samples were assessed under dynamic loads to assess their piezoelectric sensing capacity. Figure 3a illustrates the experimental setup. Compressive loads were applied to each cement or concrete sample using MTS 809 Axial/Torsional Test Systems, Inc., Eden Prairie, MN, USA. The piezoelectric responses were measured by a digital bench multi-meter (BK 5492B, B&K Precision Inc., Phoenix, AZ, USA). The samples were subjected to dynamic loading, as shown in Figure 3b, with an average load of 1912 N and a range from 166 to 2078 N in 10–12 loading cycles. The responses are presented in µV (×10^6^) corresponding to the applied loading. The frequency of the loading was set to 0.1 Hz. All the samples were tested at room temperature.

## 3. Results

### 3.1. Direct Mixing Method

Many previous studies investigating piezoelectric properties of CNT-modified smart cementitious materials used pure cement as the matrix material [32,41,42]. Figure 4a–c presents the piezoelectric responses (µV) from the samples of smart cement mortar prepared using the direct mixing method with 0.6, 0.5 and 0.4 W/C ratios, respectively. As shown in Figure 4, there was no resemblance in the dynamic responses to the stress levels of the cyclic loads. The piezoelectric response of cement mortars might change with external compressive load, but for every specific time, the value of the response did not show consistent sensing patterns to the corresponding compressive stress. As cement mortars with direct mixing failed to show fully functional piezoelectric effect for sensing purposes, no further smart concrete samples with cement, sand, and aggregate were fabricated.

### 3.2. CMC Surface Treatment Method

Figure 5a–f presents the piezoelectric responses (µV) of CNT-modified smart cementitious materials for cement paste and cement mortar prepared with the CMC surface treatment method but with different W/C ratios and concrete constituents. In each cyclic loading, the maximum piezoelectric response over maximum loading is defined as the piezoelectric sensitivity, which is regarded as one of the most important parameters for piezoelectric properties of self-sensing smart materials. The pattern of all the piezoelectric responses was similar to that of the compressive stress, with similar frequency, indicating a strong correlation between the variations in the dynamic responses and stress during each cyclic load. For each one-unit N change in the force, the average corresponding change in piezoelectric response was 12.95 µV/N, 4.34 µV/N, 1.61 µV/N in the CNT-modified pure cement with the W/C ratios of 0.6, 0.5, and 0.4, respectively. A similar pattern can also be observed for the smart cement mortar. A W/C ratio of 0.6 was found to be the optimal W/C ratio. Thus, the 0.6 W/C ratio was selected to fabricate the smart CMC-treated CNT-modified concrete samples, and the results are shown in Figure 6.

### 3.3. NaDDBS Surface Treatment Method

The NaDDBS surface treatment method has been proved effective in modifying CNT dispersion in cement to achieve a piezoelectric effect [37,43]. Figure 6a–c depicts the piezoelectric responses (µV) of pure cement, cement mortars and concrete with cement, sand and coarse aggregate prepared using the NaDDBS surface treatment method with the optimal W/C ratio of 0.6, as determined from Figure 5. From Figure 5, the piezoelectric responses of the concrete specimens prepared with the NaDDBS surface treatment method showed inconsistency, with significant variation between the loading and unloading process for all three concrete constituent compositions. The dynamic responses were characterized by irregular fluctuations resembling random noise and did not show clear correlation with the changes in applied compressive stress. By examining Figure 4, Figure 5 and Figure 6, it can be seen that compared with direct mixing and the NaDDBS surface treatment, the CMC surface treatment method can be an effective approach to disperse CNTs in smart cementitious materials to achieve potential self-sensing properties.

## 4. Discussion

### 4.1. Influences of Dispersion Methods

The results above are a preliminary indication of the effectiveness of different dispersion methods. In order to further illustrate the influence of dispersion methods on the piezoelectric properties of CNT-modified cementitious materials, comparisons were made among CNT-modified cementitious materials with same testing conditions except the dispersion method. Figure 7 shows the typical piezoelectric responses of CNT-modified cement mortars with a constant W/C ratio of 0.6. The piezoelectric response of CNT-modified cement mortars with CMC surface treatment followed a similar changing pattern under the dynamic loading. The piezoelectric response changed linearly with the dynamic loading, and its value was proportional to the force levels in the consecutive cyclic loading. However, for the samples made with direct mixing and NaDDBS surface treatment methods, the piezoelectric responses remained almost unchanged regardless of loading variations, which was similar to the control samples. The average piezoelectric sensitivity of the CMC surface treatment samples was found to be 12.43 µV/N. In contrast, the sensitivities of both the direct mixing and NaDDBS surface treatment samples were substantially lower. Interconnected CNTs inside the cement paste form an electric network which facilitates electron transfer [37]. The quality of the conductive network in cementitious materials is not dependent on a single factor. Matrix porosity and the presence of aggregates may result in the discontinuation of the uninterrupted electron path, and the polarization effect may complicate the correlation between changes in electrical and mechanical conditions [44,45]. With other factors being constant, a more uniform dispersion of CNTs may increase the probability of obtaining a more continuous conductive network. Thus, when sand was added into the CNT-modified cementitious materials as fine aggregates, the CNT-modified cement mortars with CMC surface treatment achieved better piezoelectric properties, whereas CNT-modified cement mortars prepared with direct mixing and NaDDBS surface treatment methods exhibited limited piezoelectricity. The effects of different dispersion methods on CNT dispersion were described in a previously published paper by the authors [46], which indicated that the CMC treatment resulted in smaller particles sizes and more uniform dispersion of CNTs compared with NaDDBS and direct mixing.

### 4.2. Influences of The Water Cement Ratios

According to the previous discussion, CNT-modified smart cementitious materials with dispersion methods using direct mixing or NaDDBS surface treatment failed to exhibit significant piezoelectricity. Therefore, the discussions regarding W/C ratios and concrete constituents were based on CNT-modified smart cementitious materials with the CMC surface treatment method only. Figure 8 shows the average loading and unloading piezoelectric sensitivity of CNT-modified pure cement paste with the CMC surface treatment method but different W/C ratios. A higher piezoelectric sensitivity, provided it falls within a reasonable range and does not bring about excessive amount of irrelevant data, can be an indication of improved piezoelectricity and self-sensing capability. When the W/C ratio was 0.4 for the pure cement samples, the average loading and unloading piezoelectric sensitivities were 1.602 and 1.489 µV/N, respectively. As the W/C ratio increased to 0.5, the piezoelectric properties of both loading and unloading were increased to 4.369 and 4.028 µV/N, respectively, with the increments reaching around 190% compared with those for the 0.4 W/C ratio. For the W/C ratio of 0.6, the loading and unloading sensitivities increased up to eight times (12.950 and 11.950 µV/N) compared with the corresponding values of the 0.4 W/C ratio samples. Increasing the W/C ratio showed significant increases in piezoelectric sensitivity of CNT-modified pure cement pastes, which is consistent with previous research findings [46].

A similar tendency was also found for the cement mortar samples. Figure 9 displays average loading and unloading piezoelectric sensitivities of CNT-modified cement mortars among different water-cement ratios. With the W/C ratio of 0.4 for cement mortars with CMC surface treatment, the average loading and unloading piezoelectric sensitivities were 1.829 and 1.667 µV/N, respectively. After the W/C ratio increased to 0.5, compared with the CNT-modified cement mortars with 0.4 W/C ratio, a dramatic increase (about 170%) was seen for both the loading and unloading piezoelectric sensitivities, with the values being 3.754 and 3.445 µV/N, respectively. When further elevating the W/C ratio to 0.6, the piezoelectric sensitivities also jumped dramatically to 9.339 and 8.595 µV/N for loading and unloading, which were more than 5 times the values of the 0.4 W/C ratio. The improvement in piezoelectric sensitivity of CNT-modified cement mortars as a function of W/C ratio may be attributed to the same mechanism as pure cement.

### 4.3. Influences of Concrete Constituents

Figure 10 presents the average loading and unloading piezoelectric sensitivities of CNT-modified smart cementitious materials with the same W/C ratio of 0.6 but different concrete constituents. It is clearly shown in the figure that compared with CNT-modified pure cement as the matrix material, the piezoelectric sensitivity reduced moderately when sand was added to the cement. The cement mortars attained loading and unloading piezo sensitivity values of 9.34 µV/N and 8.59 µV/N, respectively, which were 27.87% and 28.12% lower than those of pure cement samples, respectively. After adding the coarse aggregates into the cement mortars, the sensitivities further declined progressively to 2.76 µV/N and 2.54 µV/N, respectively, with the decreasing increments being 78.68% and 78.74%, respectively. It is not surprising that the addition of fine and coarse aggregate had a negative effect on the piezoelectric properties of CNTs due to the interruption of the continuous conductive network and the intrinsic high resistivity of aggregates. This influence was found to be highly substantial. The piezoelectric properties of CNT-modified smart concrete were excessively limited compared with pure cement. Thus, when fabricating smart or self-sensing concrete in practical applications, the existence of fine and coarse aggregates cannot be ignored, and the influence of aggregates on the sensing performance of the concrete need to be considered. In addition, as expected, the sensitivities of loading were consistently slightly larger than those of unloading under all testing conditions, due to the occurrence of modest irreversible plastic deformations generated in the loading cycles.

## 5. Conclusions and Future Work

This study investigated the influences of CNT dispersion methods, the W/C ratio, and concrete constituents on the piezoelectric sensing ability of CNT-modified smart cementitious materials. The following conclusions can be drawn based on the findings from this study:The comparison between three different dispersion methods showed that the CNT-modified cementitious materials with CMC surface treatment showed functional and significant piezoelectric responses with better consistency, whereas direct mixing and NaDDBS surface treatment failed to exhibit an obvious piezoelectricity in CNT-modified cementitious materials.With the increase in the W/C cement ratio, the piezoelectric sensitivities of both CNT-modified cementitious materials with and without sand addition were improved significantly, which was consistent with previous research findings. Specifically, this study found that by increasing the W/C ratio from 0.4 to 0.6, the piezoelectric responses can potentially increase by more than eight times.Adding aggregates significantly reduced the piezoelectric sensitivities of CNT-modified cementitious materials. Especially, for CNT-modified smart concrete with both fine and coarse aggregates, the sensing ability could decline to 78% of that of pure cement. When fabricating CNT-modified smart concrete, the influences of aggregates should be considered.

The conclusions of this paper confirm the importance of aggregates on the piezoelectric properties and self-sensing performance of CNT-modified cementitious materials. Even with optimized dispersion method and W/C ratios, CNT-modified concrete with sand and coarse aggregates ended up with a rather weak piezoelectric sensitivity. Thus, improving the piezoelectric sensitivity of CNT-modified concrete still needs further investigation.

## Figures and Tables

**Figure 1 sensors-23-02602-f001:**
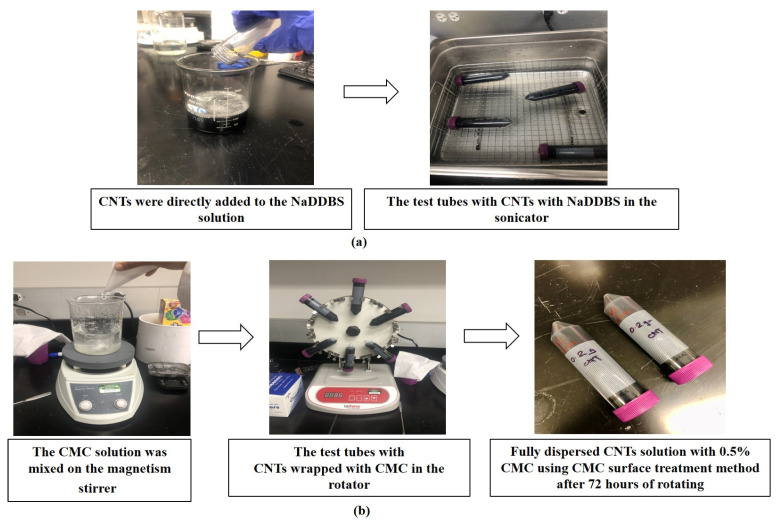
Key dispersion procedures: (**a**) NaDDBS surface treatment method; and (**b**) CMC surface treatment method.

**Figure 2 sensors-23-02602-f002:**
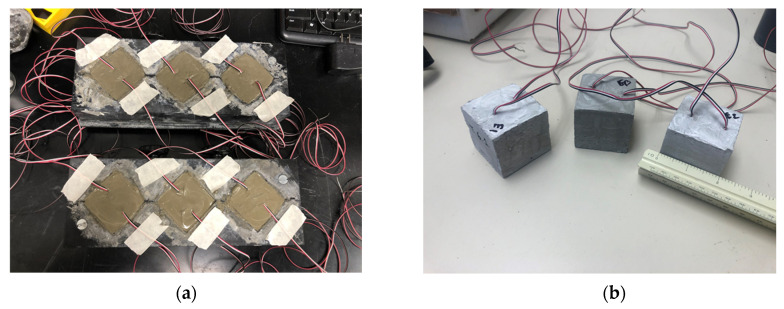
Test samples: (**a**) the cubic molds for fabricating samples; and (**b**) samples embedded with electrical wires after curing.

**Figure 3 sensors-23-02602-f003:**
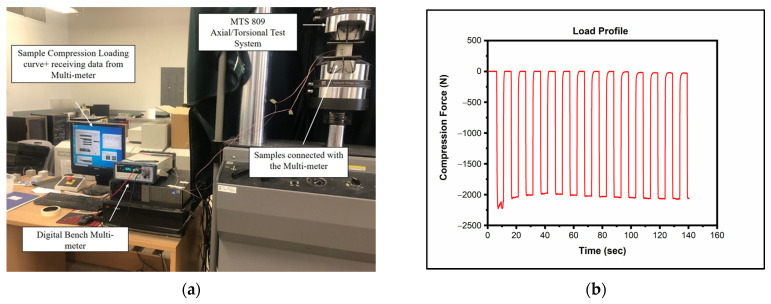
(**a**) Laboratory setup for full experimental setup. (**b**) Dynamic loading curve.

**Figure 4 sensors-23-02602-f004:**
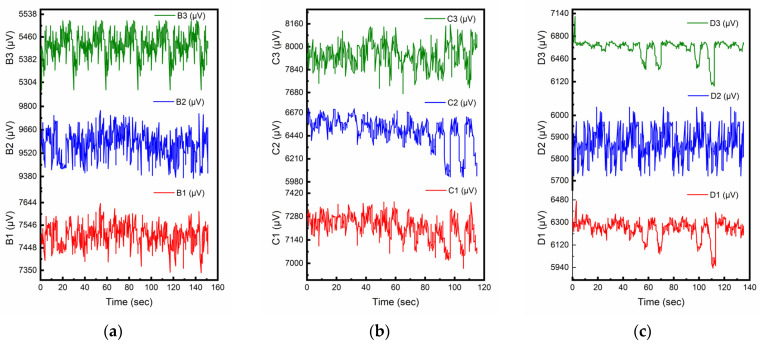
Piezoelectric responses for direct mixing method with different W/C ratios. (**a**) W/C = 0.6 (Cement:Sand = 1:2). (**b**) W/C = 0.5 (Cement:Sand = 1:2). (**c**) W/C = 0.4 (Cement:Sand = 1:2).

**Figure 5 sensors-23-02602-f005:**
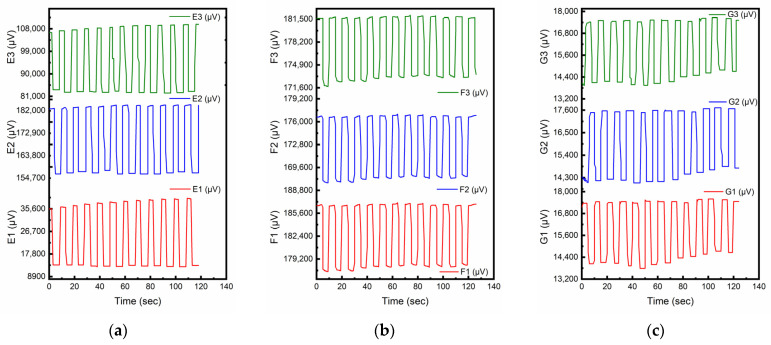
Piezoelectric responses for the CMC surface treatment method with different W/C ratios and concrete constituents. (**a**) W/C = 0.6 (Cement Only). (**b**) W/C = 0.5 (Cement Only). (**c**) W/C = 0.4 (Cement Only). (**d**) W/C = 0.6 (Cement: Sand= 1:2). (**e**) W/C = 0.5 (Cement: Sand= 1:2). (**f**) W/C = 0.4 (Cement:Sand = 1:2). (**g**) W/C =0.6 (Cement:Sand:Aggregate = 1:2:3).

**Figure 6 sensors-23-02602-f006:**
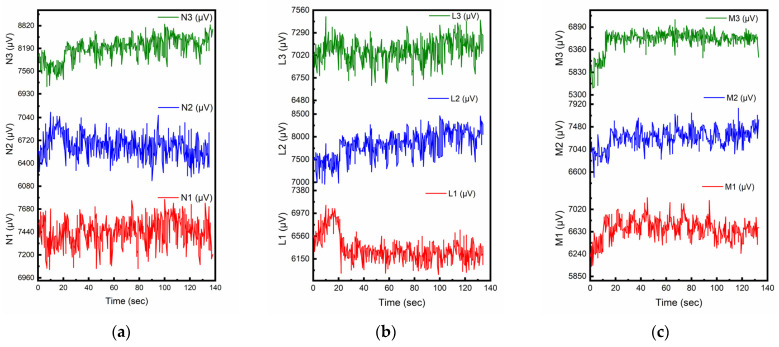
Piezoelectric responses for the NaDDBS surface treatment method with different concrete constituents. (**a**) W/C = 0.6 (Cement Only). (**b**) W/C = 0.6 (Cement:Sand= 1:2). (**c**) W/C = 0.6 (Cement:Sand:Aggregate = 1:2:3).

**Figure 7 sensors-23-02602-f007:**
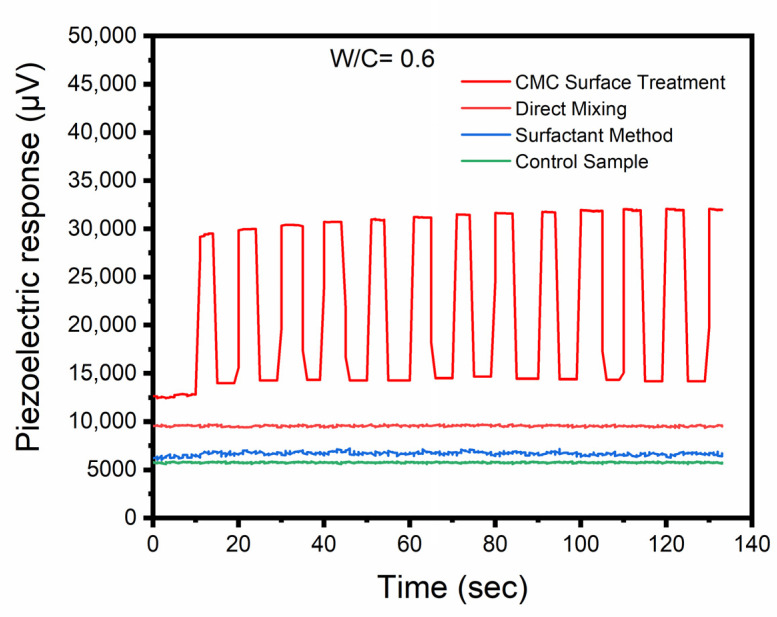
Comparisons of piezoelectric responses among the three dispersion methods with a constant W/C ratio of 0.6.

**Figure 8 sensors-23-02602-f008:**
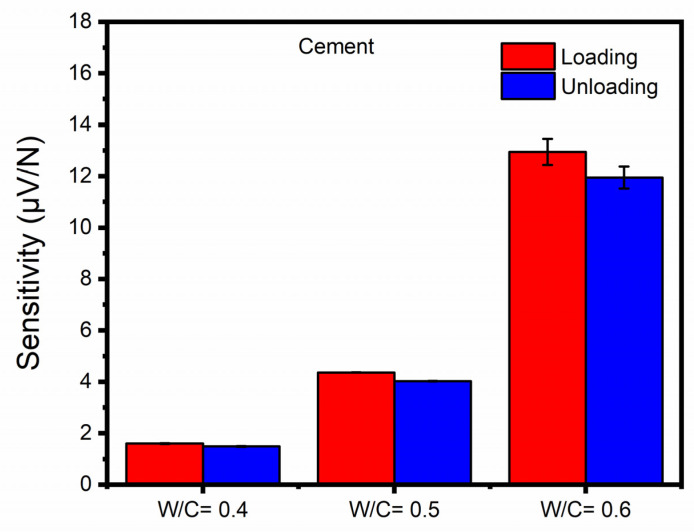
Average loading and unloading piezoelectric sensitivities of CNT-modified pure cement paste with CMC surface treatment method but different water-cement ratios.

**Figure 9 sensors-23-02602-f009:**
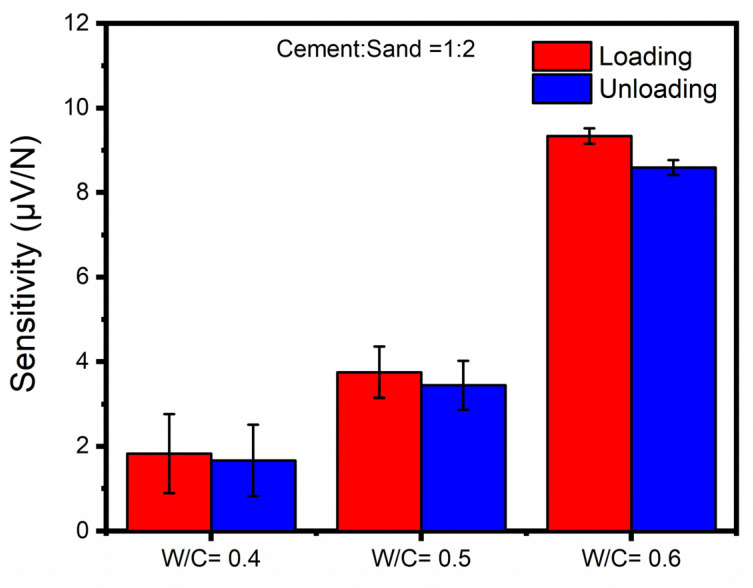
Average loading and unloading piezoelectric sensitivities of CNT-modified cement mortars with CMC surface treatment method but different water-cement ratios.

**Figure 10 sensors-23-02602-f010:**
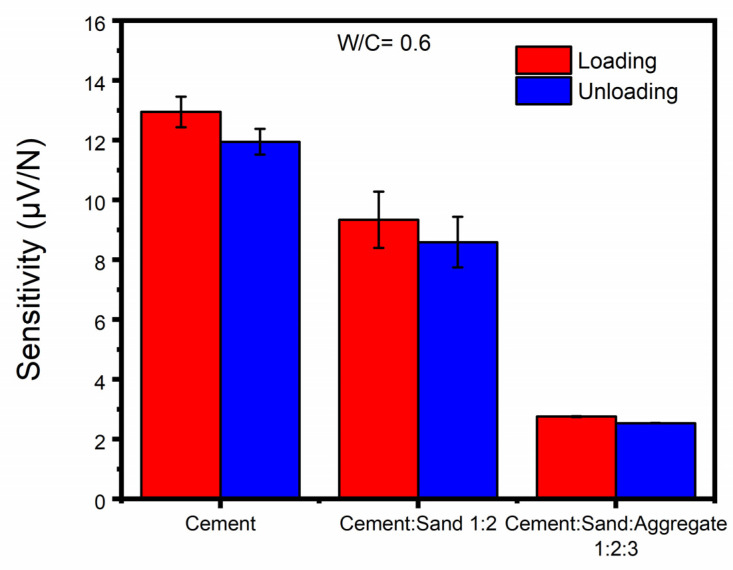
Average loading and unloading piezoelectric sensitivities of CNT-modified cementitious materials with the same W/C ratio of 0.6 but different concrete constituents.

**Table 1 sensors-23-02602-t001:** MWCNT properties.

Parameters	Specifications
Purity	>95 wt.%
Outside diameter	50–100 nm
Inside diameter	5–10 nm
Length	5–20 um
SSA	>60 m^2^/g
Ash	<1.5 wt%
Amorphous carbon	<3.0%
Electrical conductivity	>100 s/cm
Bulk density	0.28 g/cm^3^
True density	~2.1 g/cm^3^

**Table 2 sensors-23-02602-t002:** Testing sample matrix.

Dispersion Method	Group	Number of Samples	Description	W/C Ratio	Design Mix
None	A	3	Control (No CNTs and CMC)	0.6	1:2 Cement: Sand
Method 1: Direct Mixing	B	3	0.1% CNTs	0.6	1:2 Cement: Sand
	C	3	0.1% CNTs	0.5	1:2 Cement: Sand
	D	3	0.1% CNTs	0.4	1:2 Cement: Sand
Method 2: CMC Surface Treatment	E	3	0.1% CNTs + 0.5% CMC	0.6	Cement
	F	3	0.1% CNTs + 0.5% CMC	0.5	Cement
	G	3	0.1% CNTs + 0.5% CMC	0.4	Cement
	H	3	0.1% CNTs + 0.5% CMC	0.6	1:2 Cement: Sand
	I	3	0.1% CNTs + 0.5% CMC	0.5	1:2 Cement: Sand
	J	3	0.1% CNTs + 0.5% CMC	0.4	1:2 Cement: Sand
	K	3	0.1% CNTs + 0.5% CMC	Optimal W/C	1:2:3 Cement: Sand: Coarse aggregate
Method 3: NaDDBS Surface Treatment	L	3	0.1% CNTs + 0.5% NaDDBS + 0.25% deformer	Optimal W/C	1:2 Cement: Sand
	M	3	0.1% CNTs + 0.5% NaDDBS + 0.25% deformer	Optimal W/C	1:2:3 Cement: Sand: Coarse aggregate
	N	3	0.1% CNTs + 0.5% NaDDBS + 0.25% deformer	Optimal W/C	Cement

## Data Availability

Not applicable.

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
