# Peer review of "Influences of CNT Dispersion Methods, W/C Ratios, and Concrete Constituents on Piezoelectric Properties of CNT-Modified Smart Cementitious Materials"

_sensors, 2023, doi:10.3390/s23052602_

Round 1

Reviewer 1 Report

Results and discussions can be explained deeper with their specific data. 

Author Response

The authors express gratitude to the reviewer for dedicating time and efforts to offering valuable feedback on this manuscript.

Author Response

(The authors gave the same response as above.)

Reviewer 3 Report

The manuscript assessed the CNT dispersion method, w/c ratio, and the fine and coarse aggregates' influence on the piezoelectric sensing ability of CNT-modified smart cementitious materials. Overall, the manuscript is well-written and structured. However, some points need to be improved for publication:

1) “This study investigated the influences of CNT dispersion method, water/cement (W/C) ratio, and concrete ingredient on the piezoelectric properties of CNT-modified cementitious materials.” replace "ingredient" with "constituents".

2) One of the most applied methods for CNT dispersion is sonication (https://doi.org/10.1016/j.conbuildmat.2020.120237). This should be mentioned in the introduction. Also, why was this method not evaluated in the study?

3) Many surfactants can be used for CNT dispersion. However, they can negatively impact cement hydration and incorporate air, as previously reported in the literature. This also needs to be mentioned in the introduction. Why did the authors not choose to use superplasticizer additives for CNT dispersion?

4) “Another factor to highlight is that most of the existing studies 105 only added CNTs in cement paste without sand or rocks” This information is very interesting and indicates one of the gaps concerning this topic.

5) Provide the characterization of CNT (diameter, length, type of functionalization, specific surface, etc.)

6) “Type 1 Portland cement, all-purpose sand, and gravels with size of approximate 3/8 in diameter was used as matrix material, fine aggregate, and coarse aggregate, respectively.”

7) Provide the characterization of cement (particle size, composition, etc.) and fine and coarse aggregates.

8) In Figure 5 insert the graph of the composition with only cement and w/c = 0.6 to facilitate the comparison.

9) When discussing the results, it is essential for the authors to mention the effect of surfactants on CNT dispersion and how this can affect self-sensing properties. What does the literature say about the impact of the evaluated surfactants?

Author Response

(The authors gave the same response as above.)

Round 2

Reviewer 2 Report

Rewrite the sentences in lines 57, 63, 275, 331.

Line 106 – “While in practice concrete always has sand, fine and coarse aggregates in…” – Sand is a fine aggregate.

Please express the diameter of all aggregates, including sand. Use the units consistently, where the units of length correspond to each other and also correspond to the units of stress, force, etc.

“The superplasticizers usually are applied for water reduction while pursuing similar workability and high strength, which are expected to have limited influences on the piezo-electric properties.” – The sentence is unnecessary, i.e., not contributing to the topic. If I may suggest, it could be used in the previous sentence as an additional reason for not using the superplasticizers.

“…cubic concrete cubes.” – Pleonasm.

“Several factors, such as matrix porosity, polarization effect, and presence of aggregates may result in the discontinuation of the uninterrupted electron path. Better dispersed CNTs may increase the chances for a more continuous conductive network.” – The quality of dispersion is not an independent characteristic of the composite material. The authors stated that even though several factors may interrupt the network, better dispersion may increase it. Polarization effect does not affect the network physically and as such does not interrupt it. Furthermore, matrix porosity discontinues the conductivity in terms of the tunneling and ionic conductivity and better dispersion has no impact in that case. The authors should consider this topic deeper before making such statements.

“Thus, it was reasonable to conclude that the increase of piezoelectric properties as the increase of W/C ratio was valid for all CNT modified smart cementitious materials.” – It is proven and widely known fact the w/c ratio influences the piezoresistivity.

“It indicates that as the addition of fine and coarse aggregates, the piezoelectric properties of CNT modified smart concrete were rather limited compared to pure cement.” – Again, it is a known fact that aggregates increase piezoresistivity not only due to interruptions but also because of their intrinsic high resistivity.

“...it was found that the sensitivities of loading were always slightly larger than those of unloading.” – It is an expected occurrence.

Reviewer 3 Report

Accept

Author Response

The authors sincerely appreciate your valuable feedback and positive suggestion.